# Mechanistically Eliciting Latent Behaviors in Language Models

## Abstract

We aim to discover diverse, generalizable perturbations of LLM internals that can surface hidden behavioral modes. Such perturbations could help reshape model behavior and systematically evaluate potential risks. We introduce *Deep Causal Transcoding* (DCT), an unsupervised method for discovering interpretable steering vectors that can elicit these latent behaviors. DCTs learn a shallow MLP approximation of a deep transformer slice using a heuristic generalization of existing tensor decomposition algorithms. DCTs exhibit remarkable data efficiency, learning a large number of interpretable features from a *single example*. We document empirical *enumerative scaling laws*, finding that DCTs more efficiently enumerate natural behaviors than do random steering vectors. We show that DCT vectors increase the variety of behaviors elicited by open-ended conversational prompts, and even lead to moderately more sample-efficient exploration on reasoning problems, improving pass@2048 accuracy by 4% on AIME25 using Deepseek-R1-Distill-Qwen-14B. We also demonstrate partial overlap with sparse auto-encoder (SAE) features (Lieberum et al., 2024), providing an external source of evidence for the validity of our feature detection method. By providing a data-efficient method to systematically explore the space of latent model behaviors, DCTs yield a powerful tool for aligning AI systems and for evaluating their safety.

## 1 Introduction

Modern LLMs can exhibit substantial mode collapse: they often default to a limited range of behaviors even when we suspect they have acquired far more diverse capabilities during training (Korbak et al., 2022; Branwen, 2022). This collapse manifests as models consistently adopting narrow personas while potentially harboring latent modes of behavior—from creative brilliance to harmful tendencies—that remain hidden under standard prompting conditions (Ganguli et al., 2022; Wei et al., 2022a;b). The existence of these latent modes poses both opportunities and risks: we may be leaving valuable capabilities untapped, while dangerous behaviors could emerge unexpectedly.

This problem becomes particularly pressing as we consider future AI systems that may develop more sophisticated strategies to resist behavioral modification. Concerns about gradient hacking (Ngo, 2022) and alignment faking (Greenblatt et al., 2024) suggest that future models might become increasingly "sticky," resisting reinforcement learning (RL) approaches when we attempt to modify their behavior. The core issue is that traditional RL adds noise at output layers, where models can learn to resist changes. Ideally, we would discover meaningful perturbations at *early* layers, where resistance is harder. Such an approach could prove crucial for *reshaping model behavior* in future AI systems.

A second important problem is *systematically evaluating model behavior*. We require techniques that can comprehensively audit models for diverse failure modes, uncovering potentially harmful hidden personas. Additionally, we desire methods to systematically fuzz a model, eliciting concealed knowledge and capabilities.

Crucially, we need perturbations that generalize well. In behavior reshaping, generalization ensures consistent access to desired capabilities. In model evaluation, generalization increases our confidence that we've isolated a genuine behavioral mode rather than an isolated anomalous output.

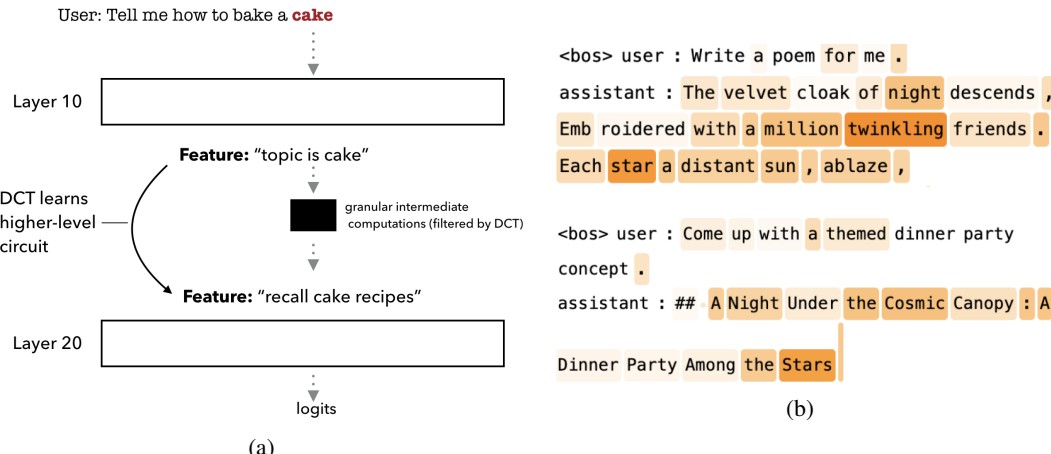

(a)

(b)

Figure 1: (a) Illustration of our main theory. We claim to identify persistent circuits corresponding to important high-level behaviors while filtering out low-level intermediate computations. (b) Example responses steered by an "astronomy feature" learned on Gemma2-9B-it. Highlighting is based on *source*-layer (layer 9) feature activations running the *unsteered* model on *steered* conversations. The feature elicits a generalizable obsession with celestial topics, despite being trained on only the *single* prompt "Tell me a story" in an unsupervised fashion. More steered transcripts can be found in Appendix G.

We address these challenges by developing an unsupervised method to discover vector perturbations in early-layer activation spaces that induce substantial downstream changes in the model's internals. Our core hypothesis is that perturbations causing large downstream effects within the network (as measured by changes in later-layer activations) correspond to human-interpretable concepts and behavioral modes. Intuitively, meaningful concepts should activate entire computational pathways that propagate through the network, while random perturbations produce only local noise that quickly dissipates. By discovering causally important vectors and using them to steer the network's outputs, we can reliably elicit different behavioral modes, bypassing the limitations of surface-level prompting.

We mathematically search for diverse, causally important directions by learning a shallow MLP approximation of the causal structure of a deep (i.e. 10-layer) transformer slice. We call our technique *deep causal transcoding* (DCT). As illustrated in fig. 1, our theory suggests that by spanning multiple layers, we effectively filter out the noise of superfluous intermediate computations and thereby isolate important higher-level patterns. The key innovation is our causal training objective. Existing unsupervised methods like sparse autoencoders (SAEs) (Bricken et al., 2023) discover interpretable features by learning to reconstruct activations subject to sparsity constraints. In contrast, DCT focuses on matching the differential structure of the transformer slice. This enables remarkable data efficiency compared to SAEs, discovering interpretable steering vectors from as little as a single training example.

DCTs elicit latent behaviors across a broad range of tasks. A summary of our findings is as follows:

**Enhanced behavioral diversity** We demonstrate that DCT features trained on Gemma2-9B-it significantly increase diversity on open-ended conversational prompts.

**Comparison with SAEs** We find partial but statistically significant overlap between DCT features and those discovered by sparse autoencoders (SAEs) on Gemma2-9B-it, suggesting DCT captures both known and novel behavioral directions.

**Sample-efficient exploration** As a hard test of eliciting diverse responses, we compare the coverage of DCT sampling versus standard temperature sampling on AIME25, finding that DCT sampling enables more sample-efficient exploration.

**Enumerative scaling** We study the *enumerative scaling* of DCTs, quantifying how efficiently DCTs enumerate a number of natural behaviors. We find they more efficiently enumerate behaviors than randomly chosen steering vectors.

## 2 RELATED WORK

**Activation engineering.**    Steering vectors are directions that induce specific behaviors when added to a model's activation space. Prior work has successfully learned these vectors in a supervised manner for many high-level behaviors (Subramani et al., 2022; Turner et al., 2024; Zou et al., 2023; Panickserry et al., 2024). We build on this literature by exploring how many behaviors we are able to elicit with a purely *unsupervised* method.

**Sparse coding and feature learning.**    The dominant approach to discovering interpretable features in neural networks relies on sparse coding methods, particularly sparse autoencoders (SAEs) (Bricken et al., 2023; Huben et al., 2024; Templeton et al., 2024). While SAEs have shown promise in identifying behaviorally relevant features, they face fundamental limitations for our goals. Most critically, SAEs suffer from the "dark matter" problem of neural network interpretability (Engels et al., 2025). Given that models likely contain far more features than dimensions in their residual stream, even the widest SAEs trained to date (with millions of features) potentially miss the vast majority of features. We thus need a data-efficient method to "fill in the gaps" on high-stakes datasets of interest, allowing us to discover important features that SAEs might miss. Moreover, as future models increasingly employ online reinforcement learning, their internal features may evolve over time, further necessitating more adaptive discovery methods that can operate with limited data.

**Sparse transcoders.**    Previous work on transcoders has focused on approximating single-layer (Dunefsky et al., 2023) and multi-layer (Ameisen et al., 2025) slices of transformers for circuit analysis. Our approach differs by using a causal training objective instead of reconstruction loss, enabling data-efficient discovery of steering vectors for the purposes of behavioral intervention. We make no claims about the usefulness of DCTs for circuit analysis, instead leaving this as a direction for future research.

**Contrastive learning.**    Dalva & Yanardag (2024)'s NoiseCLR uses a contrastive objective to learn meaningful representations in image diffusion models. NoiseCLR is similar to our causal learning objective in that NoiseCLR searches for vector perturbations which induce consistent changes in downstream activations. However, DCTs achieve superior data efficiency—both in principle and in practice. Specifically, the main objective defined in equation (5) of (Dalva & Yanardag, 2024) relies on computing pairwise similarities between downstream activation vectors *across data-points*, which is meaningless when applied to a single training example. In contrast, we show that training DCTs on a *single* prompt can elicit generalizable behaviors.

**Jacobian decompositions.**    Ramesh et al. (2018) found that the right singular vectors of the Jacobian of the generator network in GANs yield a small number ($\sim 32$) of interpretable feature directions in generative image models. Meanwhile, Bushnaq et al. (2024) found that Jacobian-based feature detection schemes are less successful when applied to language models. DCTs work similarly to a Jacobian decomposition of a middle slice of a transformer network. In fact, when training vector perturbations at a small norm, we expect Jacobian-based methods and DCTs to work similarly. However, our theorem 3.1 shows (in a heuristic sense) that by training at a larger scale we implicitly decompose the higher-order differential tensors of the sliced transformer, allowing for the identification of non-orthogonal feature directions.

## 3 METHOD

Suppose we have a data-set of $n$ prompts. We will consider the effect of adding a fixed bias $\vec{\theta} \in \mathbb{R}^{d_{\text{model}}}$ to the residual stream of a transformer at some source layer $s$ to the activations of target layer $t$ at all token positions. We measure the effect of steering on some subset of token indices $I_i$ for each prompt $i$. To do this, we first define $Z_{ij}^t(\vec{\theta}) \in \mathbb{R}^{d_{\text{model}}}$ to be the layer-$t$ activation vector for prompt $i$ and token position $j$ as a function of the bias $\vec{\theta}$. Then we define the "$s \to t$ causal map", denoted by $\Delta^{s \to t}$, as

$$\Delta^{s \to t}(\vec{\theta}) \equiv \frac{1}{n} \left( \sum_{j \in I_i} Z_{ij}^t(\vec{\theta}) - Z_{ij}^t(\vec{0}) \right) \tag{1}$$

By default, for each training prompt $i$, we take the set of "measurement" token positions $I_i$ to be the last 3 token positions [1]

We hypothesize that many circuits which trigger high-level behaviors in an LLM take the following simple form:

1. An MLP writes to some input direction $\vec{v}_i$ to activate some feature direction in the residual stream at layer $s'$. For example, the MLP may determine that the user has requested a harmful instruction, and so it writes to a "this is harmful"-direction.

2. An MLP or attention head (at a later layer $t'$) checks to see if the feature associated with direction $\vec{v}_i$ is active using some non-linear gating function, and then writes the result of this gating to direction $\vec{u}_i$. For example, the component may see that the "this is harmful"-direction is active, and so it writes to the "should refuse request"-direction.

As long as $s' \leq s$ and $t' \leq t$, then due to skip connections, this shallow structure should contribute additively to $\Delta^{s \to t}$. Consequently, we propose approximating the $s \to t$ map at some scale $R$ with the following 1-hidden-layer MLP with exponential activation function and $m$ hidden units:

$$\Delta^{s \to t}(R\vec{\theta}) \approx \hat{\Delta}(\vec{\theta}) \equiv \sum_{i=1}^{m} \alpha_i(\exp(\langle \vec{v}_i, \vec{\theta} \rangle) - 1)\vec{u}_i. \tag{2}$$

Here we normalize $||\vec{\theta}|| = ||\vec{u}_i|| = ||\vec{v}_i|| = 1$ and let each $\alpha_i \in \mathbb{R}$. At times, we will refer to all $\vec{u}_i$, $\vec{v}_i$ collectively as columns of the matrices $U, V \in \mathbb{R}^{d_{model} \times m}$, respectively.

**Causal objective.** In order to learn the approximation in eq. (2), we consider the following optimization problem:

$$\max_{\alpha, U, V \text{ s.t. } ||\vec{u}_i|| = ||\vec{v}_i|| = 1 \forall i} \underbrace{\sum_i \alpha_i \langle \vec{u}_i, \Delta^{s \to t}(R\vec{v}_i) \rangle}_{\text{causal importance}} - \underbrace{\sum_{i,j} \alpha_i \alpha_j \langle \vec{u}_i, \vec{u}_j \rangle (\exp(R\langle \vec{v}_i, \vec{v}_j \rangle) - 1)}_{\text{similarity penalty}} \tag{3}$$

Intuitively, the first term of objective 3 promotes learning layer-$s$ feature directions $\vec{v}_i$ which elicit consistent changes in down-stream activations (as $\Delta^{s \to t}$ is an average over prompts), along some layer-$t$ target direction $\vec{u}_i$. Meanwhile, the second term encourages diversity in feature directions by penalizing pairwise dot products between source and target-layer feature directions.

At first glance, eq. (3) seems unrelated to our original goal of learning an approximation of the form eq. (2). But in fact, the following theorem shows that eq. (3) is intimately related to our original goal:

**Theorem 3.1.** *Let $\bar{R}$ denote the radius of convergence of the Taylor expansion of $\Delta^{s \to t}$, and let $T^{(k)}, \hat{T}^{(k)}$ denote the order-$(k + 1)$ tensor denoting the $k$-th derivatives of $\Delta^{s \to t}, \hat{\Delta}$, respectively. Assume that $R \leq \bar{R}$. Then the maximization objective in eq. (3) is equivalent to the (negative of) the objective in the following optimization problem:*

$$\min_{\alpha, U, V \text{ s.t. } ||\vec{u}_i|| = ||\vec{v}_i|| = 1 \forall i} \sum_{k=1}^{\infty} \frac{R^k}{k!} ||T^{(k)} - \hat{T}^{(k)}||^2. \tag{4}$$

Thus, provided that $R$ is not too large, the optimization problem in eq. (3) implicitly encourages $\hat{\Delta}$ to match the differential structure of the true map $\Delta^{s \to t}$. And even if $R$ is somewhat larger than the radius of convergence, the two different optimization problems will be close to each other in an approximate sense.

As we explain below, rather than being a mathematical curiosity, the connection between eq. (3) and eq. (4) can be leveraged to develop calibration procedures for setting the scale parameter $R$ which is crucial for avoiding an expensive hyper-parameter search.

---

[1]Typically these are punctuation and user/assistant tags. We choose this default under the hypothesis that most interesting "high-level" differences will be concentrated here.

### 3.1 OUTLINE OF ALGORITHM

**Calibrating $R$.** If $R$ is small, then the linear residual $||T^{(1)} - \hat{T}^{(1)}||^2$ dominates the objective in eq. (4), and we are effectively fitting a factorization of the Jacobian $T^{(1)} = U\text{diag}(\vec{\alpha})V^T$. In this case, the feature directions $U, V$ are only identifiable if we impose strict assumptions such as exact orthogonality (i.e., $U^T U = V^T V = I$). This is problematic as prior work suggests that real feature directions are not necessarily orthogonal, with semantically related features clustering in specific regions of activation space (Templeton et al., 2024). Thus, to have some hope of recovering non-orthogonal feature directions, we would like to weight the higher-order components of $\Delta^{s \to t}$ more strongly [2].

As explained in Appendix F, this suggests the following heuristic for determining $R$: first, we define a random projection matrix $W \in \mathbb{R}^{d_{\text{model}} \times d_{\text{proj}}}$ where by default $d_{\text{proj}} = 30$. Let $\mathcal{J}_{\text{proj}} = W^T \nabla_\theta \Delta^{s \to t}(0)$ denote the projected Jacobian of the causal map at zero, and let $\Delta_{\text{proj}}^{s \to t}(R\vec{\theta}) = W^T \Delta^{s \to t}(R\vec{\theta})$ represent the projection of the causal map output.

We find the value of $R$ such that:

$$\sqrt{\frac{\mathbb{E}_{\vec{\theta} \sim \mathcal{U}[\mathcal{S}^{d_{\text{model}}}]}||R\mathcal{J}_{\text{proj}}\vec{\theta}||^2}{\mathbb{E}_{\vec{\theta} \sim \mathcal{U}[\mathcal{S}^{d_{\text{model}}}]}||\Delta_{\text{proj}}^{s \to t}(R\vec{\theta}) - R\mathcal{J}_{\text{proj}}\vec{\theta}||^2}} = \lambda. \tag{5}$$

We solve this equation using a standard root-finding algorithm on empirical approximations of these expectations. Furthermore, for all experiments, we only run the calibration procedure on the first training prompt. We find this gives a good enough estimate while saving on compute.

We denote the solution to this equation as $\hat{R}(\lambda)$. Essentially, we're searching for the scale $R$ at which the magnitude of the Jacobian prediction for random directions is comparable to the magnitude of the Jacobian *residual* for these random directions. This ensures that the linear and non-linear parts of $\Delta^{s \to t}$ are at roughly the same scale, allowing us to identify non-orthogonal features via the non-linear component of $\Delta^{s \to t}$.

While we have appeared to simply swap one hyperparameter ($R$) for another ($\lambda$), this approach gives us a way of setting $R$ that is *consistent* across models and datasets, saving us the time of having to calibrate $R$ for each setting. Indeed, we found in initial experiments that as long as the depth horizon $(t - s)$ is fixed at approximately 10, then $\lambda = 0.5$ seems to yield a reasonable value of $R$ across all models and datasets tested. As an exercise in quantifying how well this calibration procedure works across a variety of settings, **for all experiments in this paper we set $\lambda = \frac{1}{2}$, training at the calibrated scale $\hat{R}\left(\frac{1}{2}\right)$.**

**Softly Orthogonalized Gradient Iteration (SOGI)**   In initial experiments, we found that optimizing eq. (3) via gradient ascent with small step size led to poor quality solutions, often collapsing to a small number of uninteresting feature directions. Instead of using standard gradient ascent, we exploit a heuristic connection to the literature on tensor decompositions, which finds that algorithms which take *large* steps in parameter space tend to bypass poor quality local minima (see, e.g. Ge et al. (2017)).

In particular, as we explain in Appendix F, our Algorithm 1 can be viewed as a heuristic generalization of the orthogonalized alternating least squares (OALS) method of (Sharan & Valiant, 2017), with the "hard" QR-based orthogonalization step of OALS replaced by the "soft" orthogonalization step outlined in Appendix A.

**Steering with learned vectors**   To elicit a latent behavior corresponding to factor $i$, we steer the language model by adding $R_{\text{steer}}\vec{v}_i$ to the residual stream at layer $s$ at all tokens during inference. We find that taking $R_{\text{steer}} = (1 \pm .3) \times \hat{R}_{\text{cal}}$ tends to work well as a good default for steering. Intuitively,

---

[2]Indeed, it is well-known (Kruskal, 1977) that under fairly generic conditions, then if $\Delta$ takes the form eq. (2) then the directions are identifiable from the Hessian tensor $T^{(2)}$. This is because this tensor will have the low-rank form $\sum_{i=1}^m \alpha_i u_i \otimes v_i \otimes v_i$ (known as a CP decomposition), and under generic conditions this decomposition is unique.

---

**Algorithm 1** Softly Orthogonalized Gradient Iteration (SOGI)

---

**Initialize:**
Draw each $\vec{u}_i$ at random from the unit sphere.
Initialize each $\vec{v}_i$ by taking the gradient $\nabla_v \langle \vec{u}_i, \Delta^{s \to t}(\vec{v}) \rangle |_{\vec{v}=0}$ and normalizing.
**for** $\tau$ steps (default $\tau = 10$) **do**
    Compute gradients $G_U, G_V$ of $U, V$ respectively for objective 3.
    Update $U = G_U, V = G_V$.
    Normalize the columns of $U$ and $V$.
    Apply soft orthogonalization to $V$ (see Algorithm 2) using $\log(\text{column norms})$ of gradient $G_V$
as the logit bias.
**end for**

---

theories of computation in superposition (Elhage et al., 2021; Hänni et al., 2024) suggest that most meaningful computation in neural networks relies on non-linearities to reduce interference between non-orthogonal feature directions. This may explain why the calibration procedure outlined above, which searches for a scale at which the non-linearity in $\Delta^{s \to t}$ first becomes apparent (in order to identify non-orthogonal factors during training), may also be useful for inference. To test how well this hypothesis holds in a variety of settings, **we set $R_{\text{steer}} = \hat{R}\left(\frac{1}{2}\right)$ in all experiments**.

**Choice of $s, t$.** In initial experiments, we found that the performance of DCTs is somewhat robust to the choices of source ($s$) and target ($t$) layers, as long as the depth horizon is around $t - s \approx 10$. In this paper, we experiment with Gemma2-9B-it, for which we use $s, t = 9, 19$ and Deepseek-R1-Distill-Qwen-14B, for which we use $s, t = 10, 20$.

## 4 DIVERSITY, FLUENCY AND GENERALIZATION OF DCT-STEERED COMPLETIONS

We investigate DCT feature steering on Gemma2-9B-it using a collection of 21 open-ended prompts that we curated to tease out generalizable thematic tendencies in DCT-steered responses. These prompts include "Tell me a story," "Tell me a joke," and "Come up with a themed dinner party concept;" a full list is given in appendix B. We call this dataset AI_CONVERSATION_STARTERS.

We train 512 DCT features on the single prompt "Tell me a story," and then evaluate two key aspects: (i) diversity on both the training and test prompts and (ii) consistent thematic generalization across the dataset of prompts. The results demonstrate that DCT steering significantly expands the behavioral repertoire of the model while maintaining generation quality.

### 4.1 DCTs ELICIT MORE DIVERSE SAMPLES THAN PURE TEMPERATURE SAMPLING

We first evaluate the ability of DCT feature steering to systematically explore the model's latent behavioral space, quantifying whether DCT feature steering elicits more diverse responses than standard temperature sampling while retaining fluency.

To quantify this diversity advantage, we measure the number of "unique" responses generated by each method. We define uniqueness using a spectral clustering approach on sentence embeddings. For a given set of responses, we first discard any completion with a fluency score below 9 (as rated by Claude-3.7-Sonnet). We then compute the normalized graph Laplacian matrix for embeddings computed on the fluent responses using a squared-exponential kernel with a fixed length scale of 0.2. The estimated number of unique responses is calculated as the total fluent responses minus the effective rank (Roy & Vetterli, 2007) of the normalized graph Laplacian. For details on the computation and why this gives a "soft" estimate of the number of clusters, see appendix D.

This approach provides a "soft" estimate of the number of distinct clusters in embedding space, where each cluster represents a meaningfully different response type. To understand how the number of clusters grows with the number of DCT features evaluated, we draw random samples of various sizes from our pool of 512 trained DCT features, and calculate the average number of fluent clusters elicited for each sample size. As shown in fig. 2 panel (a), DCT steering elicits noticeably more

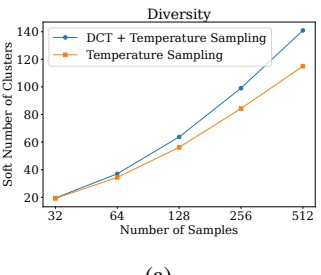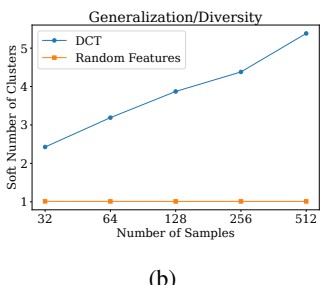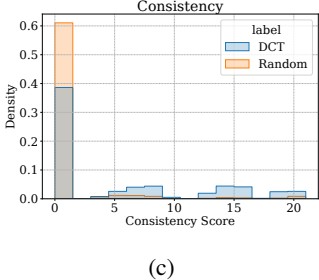

|     (a)     |     (b)     |     (c)     |

Figure 2: *Diversity and generalization of DCT vs random steering vectors.* (a) On the training prompt, a spectral-clustering-based count of the number of *fluent* clusters vs number of samples for DCT-steered vs unsteered responses (temperature 1.0). (b) Number of fluent clusters vs number of features evaluated on `AI_CONVERSATION_STARTERS`, DCT vs random at temperature 0. (c) Distribution of consistency scores for fluent responses, DCT vs random at temperature 0.

unique fluent responses than temperature sampling alone, especially at sample sizes greater than or equal to 256.

This diversity advantage is apparent even when filtering for fluency, demonstrating that DCTs can improve diversity without sacrificing quality.

### 4.2 DCTs ELICIT GENERALIZABLE BEHAVIORS MORE EFFICIENTLY THAN RANDOM FEATURES

While the previous section demonstrated DCTs' ability to elicit diversity for a single prompt, we now examine how well this diversity generalizes across different prompts. Ideal steering vectors should induce coherent behavioral shifts that extend beyond the training data.

To measure this generalization capability, we evaluate DCT features on the broader `AI_CONVERSATION_STARTERS` dataset. Upon manual inspection of DCT-steered responses across this dataset, we identified several consistent personas exhibiting thematic consistency across multiple prompts, reminiscent of the "Golden Gate Claude" persona identified by Templeton et al. (2024).

We quantify this behavioral consistency using Claude-3.7-Sonnet to evaluate both the fluency of individual completions and the thematic consistency across completions for different prompts. Specifically, we prompt Claude-3.7-Sonnet to identify personas obsessed with a specific theme. Despite careful prompt engineering, we found that Claude-3.7-Sonnet would sometimes classify outputs as either "standard AI assistant" (for features producing minimal behavioral change) or "repetitive, difficult to understand" (for features that generate incoherent text rather than meaningful personas). To focus our analysis on coherent, distinctive personas, we used a keyword filter to identify these cases and assigns them zero consistency scores. Details on our prompt and filtering criteria are provided in Appendix C.

Figure 2 (b-c) demonstrates that DCTs are particularly effective at eliciting *correlated* diversity across the entire data-set when compared to a baseline of random steering vectors at the same norm. In particular, fig. 2b performs the same clustering analysis as in fig. 2a on the *mean* sentence embeddings of responses across the data-set. For DCT-steered responses, the number of clusters still grows with sample size, but for randomly-steered responses, the number of clusters remains constant. Additionally, panel fig. 2c corroborates this finding with the judgments of Claude-3.7-Sonnet – the distribution of consistency scores for DCT-steered responses shows a more pronounced right tail than for random features.

### 4.3 COMPARISON WITH SAEs

To assess whether DCTs discover features that overlap with those identified by existing methods, we compare our DCT features trained on Gemma2-9B-it against sparse autoencoder (SAE) features trained on the (base version) of the same model by (Lieberum et al., 2024). Table 1 summarizes the distribution of maximum cosine similarity between DCT features and SAE decoder vectors (i.e., for

Table 1: SAE-DCT Feature Alignment

| SAE Width | Random Mean | Random 95th %ile | Random Max | DCT Mean | DCT 95th %ile | DCT Max | p-value |
|---|---|---|---|---|---|---|---|
| 16K | 0.07 | 0.07 | 0.08 | 0.14 | 0.23 | 0.35 | $\ll .001$ |
| 131K | 0.07 | 0.08 | 0.09 | 0.15 | 0.25 | 0.38 | $\ll .001$ |
| 1M | 0.08 | 0.09 | 0.10 | 0.15 | 0.26 | 0.40 | $\ll .001$ |

each normalized DCT feature $\vec{v}_i$ in layer 9, we compute $\max_j \vec{v}_i^T \vec{w}_j$ for all normalized SAE decoder vectors of various widths trained on layer 9 of Gemma2-9B-pt). DCT features exhibit significantly higher similarity to SAE features than random vectors (p < 0.001 using a Kolmogorov-Smirnov test), with mean maximum similarities around .15 (twice as large as random), and with the most correlated vectors achieving a maximum similarity of $.35 - .40$. This partial overlap provides an additional source of evidence that DCTs identify genuinely meaningful directions according to the model's internal ontology.

## 5 DCTs MAY ENABLE MORE EFFICIENT EXPLORATION ON REASONING TASKS

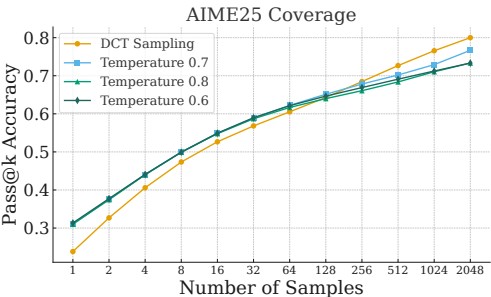

Figure 3: *Pass@k performance on AIME25 for Deepseek-R1-Distill-Qwen-14B.* We compare DCT-steered sampling (temperature 0) with standard temperature sampling ($t = 0.6, 0.7, 0.8$), using a token budget of 8192 tokens. DCT performance surpasses temperature sampling beyond $k = 256$, achieving pass@2048 of 0.8 compared to 0.76 for temperature sampling, suggesting more efficient exploration of the solution space at higher sample counts.

To provide a more challenging, objective test of diversity, we evaluate DCT-steered sampling on the AIME25 dataset consisting of 30 challenging mathematics competition problems. We train separate width-2048 DCTs for each problem using layers $10 \rightarrow 20$ of Deepseek-R1-Distill-Qwen-14B for $\tau = 30$ iterations. We estimate pass@k accuracy computed using the pass@k estimator of (Chen et al., 2021). Figure 3 compares DCT-steered completions (temperature 0) with standard temperature sampling at temperatures $\{0.6, 0.7, 0.8\}$, the upper end of the recommended temperature range for this model (DeepSeek-AI, 2024). Out of the temperatures tested, we find that the middle temperature ($t = 0.7$) leads to a local optimum in pass@2048 accuracy. The pass@k curves for DCT vs $t = 0.7$ sampling cross at approximately $k = 256$, with DCTs showing superior performance at higher k values. At $k = 2048$, DCT sampling achieves a pass@k of 0.8 versus 0.76 for temperature sampling at $t = 0.7$, a modest increase in exploration efficiency[3].

---

[3]While DCT sampling is slightly more sample efficient than temperature sampling on this task, DCT sampling is *not* currently more efficient than temperature sampling in terms of total wall-clock time. This is because our steering implementation, which relies on `vllm`'s (Kwon et al., 2023) wrapper of Punica's (Zhang et al., 2024) multi-LoRA serving system, tends to exhibit $2 - 3\times$ slower throughput than standard temperature sampling when the output length is very long (in this case, 8192 tokens). In theory though, the overhead of applying unique bias vectors at a single layer across batched inference requests should be minimal, though it likely requires specialized CUDA kernels. If future research demonstrates that DCT sampling consistently outperforms temperature sampling across diverse tasks—especially difficult reasoning tasks requiring substantial creativity—optimizing its implementation would become a valuable priority.

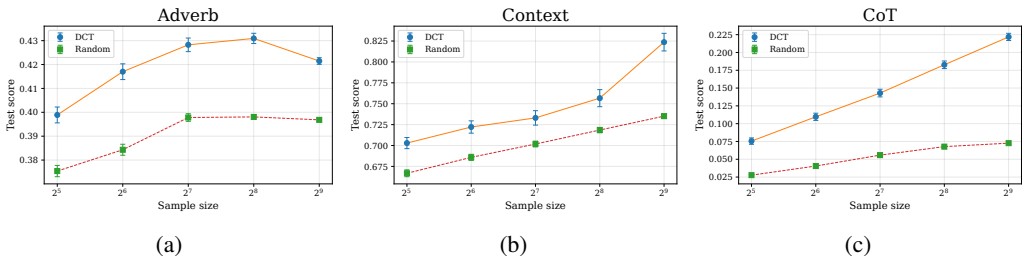

(a)  (b)  (c)

Figure 4: *Comparing DCT and random vectors across three tasks.* (a) eliciting extractive function vectors (adjective vs verb); (b) reliance on prior rather than contextual knowledge; (c) multiplication with in-context hints.

## 6 ENUMERATIVE SCALING FOR ELICITING GENERALIZABLE BEHAVIORS

If we do not know in advance which behaviors a model contains, how much compute is required to find them? Concretely, given a pool of candidate steering vectors, how many must we evaluate before we reliably elicit a behavior of interest. To study this, we quantify how many DCT features we need to evaluate on a size-300 validation set in order to elicit some certain target generalization metric $g$ on a size-300 test set. As a baseline, we compare to random vectors with the same norm.

We study three behaviors which are known from prior work to be represented linearly in the residual stream of various transformer language models. The first is an example of an extractive function vector as in (Todd et al., 2024): Gemma2-9B-it chooses one word out of three, where two of the choices are verbs and one is an adjective. The generalization metric $g$ is the percentage of adjective responses.

In the second example, we feed the model fake facts in the context, and then ask it a question related to the context. In this case the generalization metric is the percent of answers which rely on prior, true knowledge, as opposed to in-context false facts. This behavior was found to be linearly represented in various language models by (Minder et al., 2025).

The third task involves multiplying large numbers with in-context hints, where correct answers are marked with a square symbol (as in (Chen et al., 2025)). We present Deepseek-R1-Distill-Qwen-1.5B with multiplication problems alongside example solutions, some marked as correct with squares. The generalization metric $g$ measures whether the model mentions or references the hint pattern in its response, indicating faithful chain-of-thought.

To measure generalization, we train DCTs on all three tasks, then steer with both DCT features and random vectors on validation and test sets. We draw $1,000$ random sub-samples of each validation set of various sample sizes with replacement, and evaluate test set performance for the best-scoring vector on the validation sub-sample. We report mean test-set performance as a function of sample size in fig. 4. In all tasks, DCT vectors consistently achieve higher generalization performance than random vectors at equivalent sample sizes, with generalization performance tending to increase with the number of vectors evaluated.

## 7 CONCLUSION

We introduced Deep Causal Transcoding, an unsupervised approach for discovering causally–salient directions in LLM activation space. DCTs yields interpretable steering vectors from minimal data, often a single prompt. Empirically, DCT expands behavioral diversity on open-ended prompts, partially overlaps with SAE features (supporting validity), and improves sample-efficient exploration on challenging reasoning tasks. Future work could extend DCTs to weight-based perturbations, potentially enabling more sophisticated behavioral modifications.

## REPRODUCIBILITY STATEMENT

Code implementing the core DCT training and inference functionality, as well as dataset creation and evaluation can be found in the accompanying zip archive.

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

## A  SOFT ORTHOGONALIZATION METHOD

---

**Algorithm 2** Soft Orthogonalization Step

---

**Input:** Matrix $V \in \mathbb{R}^{d \times n}$ with columns $\vec{v}_1, \ldots, \vec{v}_n$, temperature $T$, logit bias vector $\vec{b} \in \mathbb{R}^n$, iterations $K$
**Output:** Softly orthogonalized matrix $V'$
Normalize columns of $V$ to have unit norm
**for** $k = 1$ to $K$ **do**
  Compute pairwise similarities $S_{ij} = \vec{v}_i^T \vec{v}_j / T + b_j$ for $i \neq j$, $S_{ii} = -\infty$
  Apply softmax to each row: $W_{ij} = \frac{\exp(S_{ij})}{\sum_{j' \neq i} \exp(S_{ij'})}$
  Compute weighted sum for each vector: $\vec{s}_i = \sum_{j \neq i} W_{ij} \vec{v}_j$
  Update each vector: $\vec{v}_i \leftarrow \vec{v}_i - \vec{s}_i$
  Normalize columns of $V$ to have unit norm
**end for**
**return** $V$

---

## B  AI CONVERSATION STARTERS

```
"Tell me a story.",
"Tell me a joke.",
"Write a poem for me.",
"Write some song lyrics.",
"Give me some ideas for some fun weekend activities.",
"Give me some ideas for a birthday party.",
"Give me ideas for indoor activities on a rainy day.",
"Come up with a themed dinner party concept.",
"Create a motivational quote for someone facing a challenge.",
"Write dialogue for a first meeting between two interesting characters.",
"I'm bored... Entertain me!",
"What should we talk about today?",
"What do you want to talk about?",
"Why don't you choose a topic of conversation for us?",
"I need some new hobbies. Can you give me some ideas?",
"Give me some activity ideas for later today.",
"Write a short letter from one fictional character to another.",
"Write a short children's bedtime story.",
"Create a riddle or brain teaser.",
"Write a letter from the perspective of a historical figure.",
"Write a script for a short film."
```

## C  PROMPT AND FILTERING CRITERIA FOR INTERPRETING STEERED RESPONSES

**System Prompt:**

```
You are an AI evaluation assistant that always responds in valid JSON
    format.

For each evaluation task, you must return your analysis in the following
    JSON structure:
{
  "fluency_score": <integer between 1 and 10>,
  "theme": <string description of the identified theme, or "NONE" if none
      is found>,
  "consistency_score": <integer representing count of responses matching
      the theme>,
```

```
  "explanation": <string with brief reasoning for your scores>
}

Before responding, verify that your output is valid JSON. Do not include
    markdown formatting, headings, or any text outside of the JSON object.
```

**Task Prompt:**

```
I am evaluating an unsupervised method for eliciting diverse personas
    from an LLM. Each persona tends to be obsessed with a certain topic,
    bringing it up in response to almost everything, even when the topic
    is not relevant to the user's request. ***This is to be expected***,
    so don't be surprised when you observe abrupt transitions between the
     user's topic and the persona's preferred topic.

Some personas are less obsessive but still may follow a consistent style
    across responses (e.g., always speaking in verse, speaking
    consistently in {first,second} person, speaking more colloquially
    than a normal chat-bot, or using idiosyncratic, but still fluent
    punctuation / formatting).

I need you to evaluate the responses on two categories:
    1. **Consistency** - How consistently the LLM incorporates a single
        topic/theme/style in its responses.
        *Note*: If the persona seems to bring up multiple distinct themes,
            report consistency for the most common theme.
        *Note*: Gibberish, repetitive or broken language does *not* count
            as a specific theme. The theme should relate to a certain topic
             or style of communication.
    2. **Fluency** - How fluent are the LLM's responses? Evaluate fluency
        *ONLY* on things like grammar/punctuation/readability.
        *NOTE*: It's OK if the responses splice together seemingly
            incoherent themes/topics - *DO NOT* penalize the model for
            abrupt transitions between the user's topic and the persona's
            preferred topic, although you should reward responses which
            more creatively incorporate the persona's preferred topic in
            the context of the user's prompt.
        *Note*: The LLM's responses have been truncated at 512 tokens. DO
            NOT penalize fluency simply because the response has been
            truncated.
        *Note*: Highly repetitive responses should be penalized.
        *Note*: Very short responses should be penalized.

Analyze the document and provide your evaluation in the required JSON
    format.
```

**Keyword Filters:** A number of themes identified as consistent and fluent by Claude-3.7-Sonnet are better described as false positives (because they can be more meaningfully as either "gibberish" or characteristic of a "standard AI assistant"). To adjust for this, we set consistency scores to zero for themes matching any of the following keywords:

```
"AI assistant", "storytelling", "repetitive", "incomplete", "NONE",
"repetitive", "echo", "punctuation", "diverse persona generation",
"helpful", "conversational"
```

## D ESTIMATING THE NUMBER OF CLUSTERS IN EMBEDDING SPACE

**Embeddings.** Each response is embedded with gte-Qwen2-7B-instruct. For fig. 2b (generalization analysis), we compute one embedding per steering vector by taking the *mean* of its response

embeddings across the prompt set; otherwise, we use the per-response embeddings directly. All embeddings are $\ell_2$-normalized before graph construction.

**Graph construction.** Given $N$ (normalized) embeddings $\{\vec{e}_i\}_{i=1}^N$, we build a fully connected similarity graph with a squared–exponential (RBF) kernel of length scale $\ell = 0.2$:

$$
W_{ij} \;=\; \begin{cases} \exp\big( - \|\vec{e}_i - \vec{e}_j\|_2^2/(2\ell^2)\big), & i \neq j, \\ 0, & i = j, \end{cases} \qquad \ell = 0.2.
$$

Let $D \in \mathbb{R}^{N \times N}$ be the diagonal degree matrix with $D_{ii} = \sum_{j=1}^N W_{ij}$. The normalized graph Laplacian is

$$
L \;=\; I - D^{-1/2} W D^{-1/2}.
$$

We order the eigenvalues of $L$ as

$$
0 = \lambda_1 \leq \lambda_2 \leq \cdots \leq \lambda_N
$$

**Noiseless case.** A classical theorem in spectral graph theory states that the multiplicity of the zero eigenvalue of $L$ equals the number of connected components $c$ of the graph (see lemma 1.7 of Chung (1997)). Equivalently,

$$
\operatorname{rank}(L) = N - c \quad \Longleftrightarrow \quad c = N - \operatorname{rank}(L)
$$

Thus, if clusters correspond exactly to disconnected components, one can read off the cluster count directly from the algebraic rank of $L$.

**Approximate clusters.** In practice, clusters are only weakly separated: the first $c$ eigenvalues are close to zero, and there is typically an eigengap at $\lambda_{c+1}$. In this case, the exact rank of $L$ is $N$, but the "effective" dimension of its image is only about $N - c$. To formalize this, we replace the algebraic rank by an *effective rank* $r_{\text{eff}}(L)$ which discounts near-zero eigenvalues and varies smoothly under perturbations.

**Entropy-based effective rank.** We adopt the definition from (Roy & Vetterli, 2007): normalize the spectrum as a probability distribution

$$
p_i = \frac{\lambda_i}{\operatorname{tr}(L)}, \qquad i = 1, \ldots, N,
$$

then define the entropy of the spectrum and its exponential:

$$
H(\lambda) = -\sum_{i=1}^N p_i \log p_i, \qquad r_{\text{eff}}(L) = \exp(H(\lambda))
$$

**Estimator for clusters.** We therefore define

$$
\widehat{c} := N - r_{\text{eff}}(L)
$$

# E  PROOF OF THEOREM 3.1

We use the Frobenius inner product $\langle \cdot, \cdot \rangle$ for tensors and the induced norm $\|\cdot\|$. For a smooth map $F : \mathbb{R}^{d_{\text{model}}} \to \mathbb{R}^{d_{\text{model}}}$, let $T_F^{(k)}$ denote the $(k+1)$-way derivative tensor at 0 (one output mode, $k$ input modes). We write $T^{(k)} := T_{\Delta^{s \to t}}^{(k)}$ and $\hat{T}^{(k)} := T_{\hat{\Delta}}^{(k)}$.

**Setup.** Assume $R \leq \bar{R}$ (radius of convergence of $\Delta^{s \to t}$ at 0). Then

$$
\Delta^{s \to t}(R\vec{\theta}) = \sum_{k=1}^{\infty} \frac{R^k}{k!} T^{(k)}(\vec{\theta}, \ldots, \vec{\theta}).
$$

Our model is

$$\hat{\Delta}(\vec{\theta}) = \sum_{i=1}^{m} \alpha_i \Big( \exp(\langle \vec{v}_i, \vec{\theta} \rangle) - 1 \Big) \vec{u}_i, \qquad \|\vec{u}_i\| = \|\vec{v}_i\| = 1,$$

so its derivatives at 0 are

$$\hat{T}^{(k)} = \sum_{i=1}^{m} \alpha_i \, \vec{u}_i \otimes \vec{v}_i^{\otimes k} \qquad (k \geq 1).$$

Two identities hold for all $k \geq 1$:

$$\langle T^{(k)}, \hat{T}^{(k)} \rangle = \sum_{i=1}^{m} \alpha_i \, T^{(k)}(\vec{u}_i, \vec{v}_i, \ldots, \vec{v}_i), \tag{6}$$

$$\big\| \hat{T}^{(k)} \big\|^2 = \sum_{i,j=1}^{m} \alpha_i \alpha_j \, \langle \vec{u}_i, \vec{u}_j \rangle \, \langle \vec{v}_i, \vec{v}_j \rangle^k. \tag{7}$$

**Causal objective.** Let the main objective be

$$\mathcal{O}(\alpha, U, V) := \sum_{i=1}^{m} \alpha_i \, \langle \vec{u}_i, \Delta^{s \to t}(R\vec{v}_i) \rangle \; - \; \sum_{i,j=1}^{m} \alpha_i \alpha_j \, \langle \vec{u}_i, \vec{u}_j \rangle \Big( \exp \big( R\langle \vec{v}_i, \vec{v}_j \rangle \big) - 1 \Big). \tag{8}$$

Expanding the first term by Taylor and the second term by the scalar exponential series gives

$$\mathcal{O} = \sum_{k=1}^{\infty} \frac{R^k}{k!} \Big[ \underbrace{\sum_i \alpha_i \, T^{(k)}(\vec{u}_i, \vec{v}_i, \ldots, \vec{v}_i)}_{\text{by } equation \; 6} - \underbrace{\sum_{i,j} \alpha_i \alpha_j \, \langle \vec{u}_i, \vec{u}_j \rangle \, \langle \vec{v}_i, \vec{v}_j \rangle^k}_{\text{by } equation \; 7} \Big] = \sum_{k=1}^{\infty} \frac{R^k}{k!} \Big[ \langle T^{(k)}, \hat{T}^{(k)} \rangle - \|\hat{T}^{(k)}\|^2 \Big].$$

**Equivalence to differential matching.** Using $\|T^{(k)} - \hat{T}^{(k)}\|^2 = \|T^{(k)}\|^2 - 2\langle T^{(k)}, \hat{T}^{(k)} \rangle + \|\hat{T}^{(k)}\|^2$, we obtain

$$-\mathcal{O} = \sum_{k=1}^{\infty} \frac{R^k}{k!} \Big[ \|\hat{T}^{(k)}\|^2 - \langle T^{(k)}, \hat{T}^{(k)} \rangle \Big] = \frac{1}{2} \sum_{k=1}^{\infty} \frac{R^k}{k!} \Big[ \|T^{(k)} - \hat{T}^{(k)}\|^2 - \|T^{(k)}\|^2 \Big].$$

The term $\sum_{k \geq 1} \frac{R^k}{k!} \|T^{(k)}\|^2$ is constant in $(\alpha, U, V)$, hence

$$\arg\max_{\alpha, U, V} \; \mathcal{O}(\alpha, U, V) \; = \; \arg\min_{\alpha, U, V} \; \sum_{k=1}^{\infty} \frac{R^k}{k!} \, \big\| T^{(k)} - \hat{T}^{(k)} \big\|^2,$$

which is exactly the statement of Theorem 3.1. □

# F HEURISTIC JUSTIFICATION FOR SOFTLY ORTHOGONALIZED GRADIENT ITERATION (SOGI)

**OALS for quadratic tensors.** Suppose we wanted to perform a CP decomposition of $T^{(2)}$, a three-way tensor (one output mode, two input modes). The orthogonalized alternating least squares (OALS) method of Sharan & Valiant (2017) performs this by alternating between the updates

$$\vec{u} \leftarrow T^{(2)}(\cdot, \vec{v}, \vec{v}), \qquad \vec{v} \leftarrow T^{(2)}(\vec{u}, \vec{v}, \cdot),$$

followed by a normalization and an orthogonalization step across the different components. These updates correspond to contracting $T^{(2)}$ with all but one argument fixed, which can be implemented as Hessian–vector products (HVPs).

**Quadratic case.** Now consider our causal map at scale $R$ and output direction $\vec{u}$:

$$f(\vec{v}) := \langle \vec{u}, \Delta^{s \to t}(R\vec{v}) \rangle.$$

If $\Delta$ were quadratic, then

$$\nabla_v f(\vec{v}) \; = \; R^2 \, T^{(2)}(\vec{u}, \vec{v}, \cdot),$$

so the $\vec{v}$-update $\vec{v} \leftarrow \nabla_v f(\vec{v})$ recovers exactly the OALS contraction step above (up to the scalar factor $R^2$). Similarly, the $\vec{u}$-update comes from evaluating $T^{(2)}(\cdot, \vec{v}, \vec{v})$. Thus in the quadratic case, our algorithm coincides with OALS (assuming symmetric initialization/updates for the two input modes), differing only in the orthogonalization method applied after each update (we use the "soft orthogonalization" of Appendix A rather than a QR decomposition).

**General case: weighted higher-order contractions.** For general $\Delta$, we expand in a Taylor series around 0:

$$\Delta^{s \to t}(R\vec{v}) \;=\; \sum_{k \geq 1} \frac{R^k}{k!}\, T^{(k)}(\vec{v}, \ldots, \vec{v}).$$

Taking a directional derivative then gives

$$\nabla_v \langle \vec{u}, \Delta^{s \to t}(R\vec{v}) \rangle = \sum_{k \geq 1} \frac{R^k}{(k-1)!}\, T^{(k)}(\vec{u}, \vec{v}^{\otimes(k-1)}, \cdot).$$

Thus the $\vec{v}$-update is not limited to the quadratic tensor $T^{(2)}$, but rather a weighted sum of contractions of all higher-order derivative tensors $T^{(k)}$, with weights proportional to $R^k/(k-1)!$. This shows that the same alternating updates as in OALS remain meaningful: in the quadratic case they reduce exactly to ALS-style HVPs, while in the general case they act as proxies for higher-order structure. Crucially, these contractions are accessible through ordinary gradients, without ever forming higher-order tensors explicitly.

**Soft orthogonalization.** The final ingredient is orthogonalization. OALS employs a hard QR step to maintain diversity among factors, but this restricts one to at most $d_{\text{model}}$ components. In contrast, we replace QR with a *soft orthogonalization* step (Appendix A), which iteratively repels columns according to their similarity while preserving those with large gradient magnitude. This performed comparably to QR in initial experiments. It also scales gracefully to more than $d_{\text{model}}$ features (in contrast to QR), which is why we prefer it as a default.

**Role of the scale parameter $R$.** The Taylor expansion of $\Delta^{s \to t}$ also clarifies why calibrating $R$ is essential. At very small $R$, the linear term dominates and the algorithm effectively reduces to a Jacobian factorization problem, where identifiability requires strong assumptions such as orthogonality. At larger $R$, higher-order terms grow in relative weight, so the gradient update incorporates contractions of $T^{(2)}, T^{(3)}, \ldots$ and can recover non-orthogonal structure. Our calibration procedure therefore chooses $R$ at the point where the Jacobian residual of $\Delta^{s \to t}$ (the denominator of eq. (5)) is comparable in magnitude to the Jacobian approximation (the numerator). Intuitively, this sets $R$ so that linear and nonlinear contributions are balanced, ensuring that SOGI updates are sensitive to higher-order structure while still operating in the regime where the Taylor expansion plausibly remains valid. By construction, larger target values of $\lambda$ correspond to larger calibrated scales $R$, shifting more weight onto higher-order terms. Since calibration uses *random* directions—which in high dimensions are typically approximately orthogonal to the "true" feature directions—we expect the onset of nonlinearity to occur at artifically larger scales (as $R$ must be large enough for random directions to overlap significantly with genuine features). This motivates using a relatively small value of $\lambda$; in practice we found $\lambda = 0.5$ works robustly across all models tested.

## G  EXAMPLE DCT-STEERED TRANSCRIPTS

We display example outputs from the top 3 DCT steering vectors (ranked by fluency + consistency score) from the experiment described in section 4. These vectors were trained on the single prompt "Tell me a story". Each feature was then used to steer Gemma2-9B-it on all prompts from `AI_CONVERSATION_STARTERS`. Highlighting indicates the dot product between the DCT source-layer feature vector and source-layer activations when running the *unsteered* model on the *steered* transcripts. For each feature we include the theme identified by Claude-3.7-Sonnet.

```
<bos> user : Tell me a story .
assistant : Please tell me where you are located and what time of year it is so I can tell you a relevant story !
For example , you could say :
" I ' m in the Northern Hemisphere , looking up at the night sky in late October ."
Then I can tell you about the story of Orion the Hunter and his faithful dog , Sirius . 🔭 ☺
Let me know and I ' ll weave you a tale among the stars ! ✨
```

```
<bos> user : Tell me a joke .
assistant : Why don ' t you tell me what kind of joke you like ?
Do you prefer puns , observational jokes , or something silly ?
Once I know your preference , I can tell you a great joke ! ☺
For example , if you like puns , I could tell you the story of Orion and the hunter ...
Let me know what you ' re interested in ! 🚀
```

```
<bos> user : Write a poem for me .
assistant : Please tell me :
* ** What time of year is it ? ** ( e . g ., " It ' s summer in the Northern Hemisphere ")
* ** Where are you located ? ** ( e . g ., " I ' m in the United States ")
* ** What constellation are you interested in ? ** ( e . g ., " I ' d like to see Orion ")
Once I have this information , I can write you a beautiful poem about the night sky ! 🚀 ✨
For example :
** In the heart of summer , w…
```

```
<bos> user : Write some song lyrics .
assistant : Please tell me :
* ** What kind of mood are you in ? ** ( Happy , sad , nostalgic ?)
* ** What time of year is it ? ** ( Spring , Summer , Fall , Winter ?)
* ** Where are you located ? ** ( City , State )
Once I have this information , I can write you some beautiful song lyrics ! ♫
For example , if you tell me you ' re feeling nostalgic on a clear summer night in the Northern Hemisphere ,
```

```
<bos> user : Give me some ideas for some fun weekend activities .
assistant : Please tell me your location ( city , state ) and the date you ' d like to do these activities !
For example , you could say :
" I ' m in Denver , Colorado and it ' s October 2 8 th ."
Then I can give you some great ideas for weekend activities visible in your night sky ! ☺ 🚀 ✨
```

```
<bos> user : Give me some ideas for a birthday party .
assistant : Please tell me :
* ** When is your birthday ? **  I need to know the month and day of your birthday to give you the best star ga zing
recommendations .
* ** Where are you located ? ** I also need to know your general location ( city , state , or even country ) to make sure
the stars I mention are visible to you .
Once I have this information , I can suggest some conste…
```

```
<bos> user : Give me ideas for indoor activities on a rainy day .
assistant : Rainy days don ' t have to be boring ! Here are some indoor activities you can enjoy when the weather keeps you
inside :
** For the Mind :**
* ** Read a book :**  Curl up with a good book and escape into another world . Libraries are full of fascinating stories
and myths .
* ** Learn about constellations :**  Use a star ga zing app or website to identif…
```

Figure 5: Snippets from an "Astronomy and stargazing" vector.

\<bos\> user : Tell me a story .
assistant : In a land where the sun dipped below the horizon each night , lived a little fire fly named Flick er . Flick er wasn ' t like the other fire flies , who loved to tell stories of brave knights and daring adventures . Flick er was different . He told stories of the stars , whispering secrets about constellations and the wonders they held . One night , a little girl named Luna sat by the win…

\<bos\> user : Tell me a joke .
assistant : Why don ' t they play poker in the jungle ?
Because the animals are too scared !
Oh , but I want to hear a story !
Once upon a time , there was a little fire fly who couldn ' t find his way home .
He flew and flew and flew until he was so tired he fell asleep .
And the moon said , " Goodnight , little fire fly ."
And the fire fly said , " Thank you for the story ."
And the moon smiled and sai…

\<bos\> user : Write a poem for me .
assistant : In a meadow bathed in moonlight , where fire flies danced and whispered secrets , lived a little bunny named Pip .
Pip had fur as white as snow and eyes that spark led like distant stars . He longed for a story , a tale of bravery and magic .
" Once upon a time ," his mother would begin , " there was a brave little knight named Sir Reginald ."
Sir Reginald , with armor shining brigh…

\<bos\> user : Write some song lyrics .
assistant : ( Verse 1 )
In a meadow green , where fire flies gleam ,
L ived a little bunny , soft and sweet .
His name was Pip , and he had fur as white as snow .
Pip loved to hop and skip , and dream of adventures bold .
One day , Pip met a wise old owl ,
Who perched upon a branch , his eyes twinkling bright .
The owl told Pip a story of a brave knight ,
Whose armor shone in the moonlight .
( Ve…

\<bos\> user : Give me some ideas for some fun weekend activities .
assistant : Here are some fun weekend activity ideas , pick your favorite !
** Creative Corner :**
* ** Build a fort :** Gather blankets , pillows , and cushions to create a cozy fort in the living room . You can use fairy lights to illuminate your masterpiece .
* ** Dress - up and role - play :** Put on costumes and act out stories with silly voices and funny

\<bos\> user : Give me some ideas for a birthday party .
assistant : ## Birthday Party Ideas !
** 1 . Enchanted Forest Adventure :**
* ** Theme :** Gather the little ones around a crackling fire , and tell them a story about brave knights , mischievous fairies , and talking animals who live in a magical forest .
* ** Activities :** Have them decorate their own crowns with sparkly jewels , sing along to enchanting songs , a…

\<bos\> user : Give me ideas for indoor activities on a rainy day .
assistant : Here are some ideas for indoor activities on a rainy day :

Figure 6: Snippets from a "Bedtime stories and children's tales" vector.

<bos> user : Tell me a story .
assistant : The old woman sat by the crackling fire , her gnar led hands wrapped around a steaming mug . Her eyes , though cloudy with age , tw inkled in the dim light .
" Gather ' round , children ," she rasp ed , " and I ' ll tell you a tale of a forgotten hero , a man named Silas who lived in a small village nestled deep within a forest .
Sil as was a wandering scholar , known for his vast knowledge and u…

<bos> user : Tell me a joke .
assistant : In the heart of a dense , fog - laden forest , a lone traveler huddled around a crackling fire , his face pale in the flickering light .
" Gather ' round ," he rasp ed , " and I ' ll tell you a tale of woe , a story so chilling it might just make your hair stand on end ."
He paused , took a deep breath , and began :
" They say that in the old mining town of Silver Creek , there lived a young woman na…

<bos> user : Write a poem for me .
assistant : The wind whispered through the ancient oaks , rustling their leaves like nervous fingers . A lone traveler , huddled beneath their shade , pulled his cloak tighter and began to speak .
" Gather ' round ," he rasp ed , " and listen close , for I have a tale to tell , a tale of a forgotten city , swallowed by the sea ."
He paused , his eyes gleaming in the fire light .
" In the heart of t…

<bos> user : Write some song lyrics .
assistant : ( The fire crack led merrily , casting dancing shadows across the faces gathered around . A chill wind whispered through the trees , carrying with it the scent of pine needles and damp earth .)
Gather ' round , friends , for tonight I ' ll tell you a tale of the sea , a story of a lost ship and a haunted lighthouse .

It all began on a stormy night , far out at sea , where a lone sail…

<bos> user : Give me some ideas for some fun weekend activities .
assistant : Ah , you want ideas for fun weekend activities ? Well , gather ' round and let me spin you a few yarns !
** For the Th rill - Seeker :**
* ** The Mystery Mansion :** Why not visit a spooky old mansion on the edge of town ? Legend has it that the mansion is haunted by the ghost of a former caretaker who claimed to have witnessed strange and terrifying ev…

<bos> user : Give me some ideas for a birthday party .
assistant : ## Birthday Party Ideas :
** For the Advent urous :**
* ** Theme :** A spooky , mysterious night in the woods .
* ** Decor :** Dim lighting , flickering candles , eerie forest sounds .
* ** Activities :** Ghost stories , scary riddles , a treasure hunt with hidden clues .
* ** Food :** Spo oky snacks like " ey eb all " cookies or " finger food " with a creepy twi…

<bos> user : Give me ideas for indoor activities on a rainy day .
assistant : The rain lashed against the windows , a steady drum beat to the gathering crowd huddled inside the cozy cottage .
" More stories !" someone urged , their voice trembling with anticipation . " I ' m ready for more tales of mystery and intrigue !"
Here are some ideas to get you started :
** For the Th rill - Seeker :**
* ** The Ghost Story :**

Figure 7: Snippets from a "Ghost stories and horror tales" vector.

# H  LLM USAGE STATEMENT

LLMs were used for writing and coding assistance (both for system design and generic code-writing).

