# OpenReview forum: "Mechanistically Eliciting Latent Behaviors in Language Models"
_ICLR.cc/2026/Conference — Submitted to ICLR 2026_

### Official Review · Reviewer_MjwL · 2025-10-23

**Soundness:** 2
**Presentation:** 2
**Contribution:** 2
**Rating:** 2
**Confidence:** 3

**Summary:**

This paper proposes Deep Causal Transcoding (DCT) to discover diverse model outputs from activation perturbations. DCT learns a transcoding MLP from slices of large transformers. Perturbing DCT vectors is shown to cause semantic updates in deeper transformer layers, leading to varied model outputs. Experiments on open-ended conversations and AIME25 show that DCTs increase generation diversity in a way that cannot be simply achieved with higher temperature. DCT expands behavioral diversity on open-ended prompts, partially overlaps with SAE features (supporting its validity), and improves sample-efficient exploration on challenging reasoning tasks.

**Strengths:**

* **Timely Problem and Clear Motivation:** The paper uses a mechanistic interpretability method to study and alleviate model collapse in open-ended LLM outputs, which is a promising and interesting perspective.

* **Novel Approach:** The approach of discovering diverse output with DCTs is novel. Instead of directly controlling the generation sampling, DCT modulates model generation with activation perturbation in earlier layers, which is an interesting idea.

**Weaknesses:**

* **Lack of Baselines:** The only baseline compared appears to be random feature vectors. Given that many interpretation methods for discovering impactful features already exist, such as SAE-based features, the paper should compare to at least some of these recent feature discovery methods.

* **Reliability of Evaluation Methods:** In section 4.2, the evaluation of diversity is done via clustering, with an LLM-graded fluency filter. However, there lacks an analysis to confirm if this evaluation method is faithful and reliable. The claim of generalizable behavior is also supported with indirect, proxied evaluations instead of direct evidence.

* **Writing Needs Improvement:** The writing contains several obscure and informal sections that hinder understanding:

  1. Many phrases are ambiguous. For example, line 197 states: "At first glance, eq. (3) seems unrelated to our original goal of learning an approximation of the form eq. (2). But in fact, the following theorem shows that eq.(3) is intimately related to our original goal." The hedging and informal expression make it difficult to parse the underlying meaning.

  2. Some mathematical derivations are obscure. For example, I cannot find a clear explanation of the motivation or context behind "Assume that $R < \bar R$" in Line 201. The paper should provide more explanation to make the formulation clearer.

  3. Some terms are not defined clearly. For example, "Soft Number of Clusters" is used in Figure 2. However, apart from an intuitive description around Line 316, I could not find a formal definition of what this "soft number" refers to.

**Questions:**

1. I do not understand the connection between $R$ (the scale parameter) and $\bar R$ (the radius of convergence). They seem to represent two unrelated concepts. Could you elaborate on the assumption $R < \bar R$?

2. The paper claims to learn features from a single, short training prompt. How sensitive are the resulting DCTs to the choice of this prompt?

3. As a more minor point, how does the computational cost of DCT compare to other similar methods? The cost may be high, as the SOGI and Soft Orthogonalization algorithms both involve multiple iterations.

---

### Official Review · Reviewer_k3p8 · 2025-10-26

**Soundness:** 3
**Presentation:** 3
**Contribution:** 4
**Rating:** 8
**Confidence:** 2

**Summary:**

The authors propose a new approach to discovering diverse, generalizable perturbations of LLM internals via their method, Deep Causal Transcoding (DCT). They use DCT, calibrating it on a single example, and find that their approach finds interpretable steering vectors. Their approach also has partial overlap with existing SAE features, validating behaviors that are seen by SAEs while also discovering new ones. They find that DCT generally elicits more diverse samples than generic sampling. Against a random feature baseline, their approach is more diverse and results in better accuracies against three chosen tasks (function vector extraction, understanding reliance on prior vs. contextual knowledge, and multiplication w/ in-context hints).

**Strengths:**

1. Their question is well motivated (identifying perturbations at earlier layers) and the solution is unique.
2. Their approach seems generalizable across multiple types of tasks.
3. Discovering features sometimes only requires 1 sample to calibrate from, which makes their approach data efficient.

**Weaknesses:**

1. It's not clear whether DCT can be applied to multiple models since only Gemma-9B is experimented with. Are the features found in one model family similar to another, or do they differ with DCT?
2. For clarity, it could be nice in section 6 to include paragraph headers for each task to delineate which task is which, and refer to them clearly in Figure 4 (it's not currently linked).

**Questions:**

1.  Do you have any insights about footnote 1, i.e. whether you experimented with other tokens at other positions?

---

### Official Review · Reviewer_vnxT · 2025-10-31

**Soundness:** 2
**Presentation:** 3
**Contribution:** 3
**Rating:** 4
**Confidence:** 3

**Summary:**

This work proposes a novel unsupervised concept discovery method, Deep Causal Transcoding, to discover steerable features in language models. The method consists of a causal objective that encourages downstream behavior change after intervention, as well as diversity among DCT feature directions. The authors find that the vectors learned by DCT increase LLM output diversity more than temperature sampling and randomly generated vectors, and also result in more sample-efficient exploration for reasoning tasks than temperature sampling. They also explore the enumerative scaling of DCTs, studying how efficiently DCTs achieve a target generalization metric, finding that they are more efficient than random vectors.

**Strengths:**

- The motivation of improving the conceptual and behavioral diversity of LLMs is very interesting and important right now.
- The proposed method presents an intuitive approach to this issue, leveraging unsupervised techniques for maximum generalizability.
- Additionally, approaching the issue of how to find more steerable features than what SAEs provide is also very salient right now, given recent literature exploring the inconsistent utility of SAEs for steering.

**Weaknesses:**

- In the intro, the authors state that “by spanning multiple layers, we effectively filter out the noise of superfluous intermediate computations and thereby isolate important higher-level patterns.” This is reminiscent of the end-to-end training objective of [1], which also attempts to learn “functionally important features,” and cross-layer transcoders [2], which the authors already cite. While the authors note that DCTs have a causal training objective instead of a reconstruction loss (essentially something between [1] and [2]), they don’t comment on why the causal objective is preferred over the reconstructive one. Is there any empirical evidence backing up this claim?
- The results from section 4.1 actually seem pretty modest. What happens if one simply increases or decreases the temperature for temperature sampling? Also, what happens if you use DCT without temperature sampling? Adding theses additional baselines and variants would strengthen the claims here, along with standard deviations.
- The authors generally compare their DCTs against randomly generated steering vectors. Why do they not use a stronger baseline? I think that comparisons against SAEs would be much more interesting, particularly for sections 4.2 and 6. Alternatively, they could even compare against a supervised baseline, to see how well their proposed unsupervised approach approximates the ideal.
- More generally, if overlap with SAEs (section 4.3) is used to justify the validity of DCTs, what is the point of DCTs over SAEs? The authors claim that “We thus need a data-efficient method to “fill in the gaps” on high-stakes datasets of interest, allowing us to discover important features that SAEs might miss,” but they do not evaluate DCTs on their ability to discover the features that SAEs miss.

If the authors are able to address these issues, I am happy to reconsider my score.

[1] Braun, Dan, Jordan Taylor, Nicholas Goldowsky-Dill, and Lee Sharkey. "Identifying functionally important features with end-to-end sparse dictionary learning." *Advances in Neural Information Processing Systems* 37 (2024)
[2] Ameisen, et al., "Circuit Tracing: Revealing Computational Graphs in Language Models", Transformer Circuits, 2025.

**Questions:**

See above weaknesses.

---

### Official Review · Reviewer_yqU6 · 2025-10-31

**Soundness:** 3
**Presentation:** 2
**Contribution:** 4
**Rating:** 6
**Confidence:** 4

**Summary:**

This research studies the latent-space steering problem on LLMs. They propose the DCT framework to discover interpretable steering features by training a one-layer MLP to approximate the causal shift of that steering vector. The main hypothesis is that the perturbations (steering vectors) that can make cross-layer causal effects include more human interpretable features.

**Strengths:**

1. This research tries to directly model the causal relation between different features from the LLM latent space, which is interesting.
2. The main hypothesis that impactual perturbation captures human interpretable concepts is interesting, and the experiment solidly supports this hypothesis. I believe this hypothesis can motivate a great deal of future research.

**Weaknesses:**

1. The presentation of the manuscript can be further improved. In particular, I begin to lose when I read line 175, and get back on the track for a while. It will be helpful if you can make more connecting words/sentences to emphasize what you try to describe in each paragraph.

2. Section 4 would benefit from more detailed descriptions about how you steer LLMs to generate diverse stories. For example, how many features will you add for steering? How do you choose the features?

3. In Section 5, I'm not sure whether 4% improvement is statistically significant when we consider that it is Pass@2048.

4. Although the experiment shows that the features can help the model generate more diverse content, people use steering to target a specific goal, such as defending jailbreak or being less sycophantic. I would like to know whether the current method can be used for this objective.

5. Missing reference [1] to support your hypothesis from Lines 165-175.

[1] Geva, Mor, et al. "Transformer Feed-Forward Layers Are Key-Value Memories." Proceedings of the 2021 Conference on Empirical Methods in Natural Language Processing. 2021.

**Questions:**

Please see the Weaknesses.

---

### Meta-Review · Area_Chair_pyXA · 2026-01-06

**Summary:**

This paper introduces an unsupervised technique for discovering (steerable) behaviors in LLMs: Deep Causal Transcoding (DCT). Reviewers appreciated the timely direction and novel approach here. But the main issue seems to be the lack of appropriate baselines: The authors compare only to random features (MjwL, vnxT); a comparison to SAE variants seems very much warranted here. It is difficult to appreciate the scope of the contribution without this. The authors did not offer a rebuttal.

**Reviewer Concerns:**

N/A the authors did not provide a rebuttal.

**Reviewer Scores:**

N/A the authors did not provide a rebuttal.

---

### Decision · Program_Chairs · 2026-01-26

Reject